# Accessible LAMP-Enabled Rapid Test (ALERT) for Detecting SARS-CoV-2

**DOI:** 10.3390/v13050742

**Published:** 2021-04-23

**Authors:** Ali Bektaş, Michael F. Covington, Guy Aidelberg, Anibal Arce, Tamara Matute, Isaac Núñez, Julia Walsh, David Boutboul, Constance Delaugerre, Ariel B. Lindner, Fernán Federici, Anitha D. Jayaprakash

**Affiliations:** 1Oakland Genomics Center, 355 30th Street, Oakland, CA 94609, USA; mfcovington@gmail.com (M.F.C.); 2Amaryllis Nucleics, 355 30th Street, Oakland, CA 94609, USA; 3Center for Research and Interdisciplinarity (CRI), Université de Paris, INSERM U1284, F-75006 Paris, France; guyaidelberg@gmail.com (G.A.); ariel.lindner@cri-paris.org (A.B.L.); 4Institute for Biological and Medical Engineering, Schools of Engineering, Biology and Medicine, Pontificia Universidad Católica de Chile, Santiago 7820244, Chile; elanibal@gmail.com (A.A.); tfmatute@uc.cl (T.M.); innunez@uc.cl (I.N.); ffederici@bio.puc.cl (F.F.); 5FONDAP Center for Genome Regulation, ANID–Millennium Science Initiative Program–Millennium Institute for Integrative Biology (iBIO), Santiago 8331150, Chile; 6School of Public Health, University of California Berkeley, Berkeley, CA 94720, USA; jwalsh@berkeley.edu; 7Clinical Immunology Department, U976 HIPI, Hôpital Saint Louis, Université de Paris, F-75006 Paris, France; david.boutboul@aphp.fr (D.B.); constance.delaugerre@aphp.fr (C.D.); 8Girihlet Inc., 355 30th Street, Oakland, CA 94609, USA

**Keywords:** RT-LAMP, point-of-care, biodetection, SARS-CoV-2

## Abstract

The coronavirus disease 2019 (COVID-19) pandemic has highlighted bottlenecks in large-scale, frequent testing of populations for infections. Polymerase chain reaction (PCR)-based diagnostic tests are expensive, reliant on centralized labs, can take days to deliver results, and are prone to backlogs and supply shortages. Antigen tests that bind and detect the surface proteins of a virus are rapid and scalable but suffer from high false negative rates. To address this problem, an inexpensive, simple, and robust 60-minute do-it-yourself (DIY) workflow to detect viral RNA from nasal swabs or saliva with high sensitivity (0.1 to 2 viral particles/μL) and specificity (>97% true negative rate) utilizing reverse transcription loop-mediated isothermal amplification (RT-LAMP) was developed. ALERT (Accessible LAMP-Enabled Rapid Test) incorporates the following features: (1) increased shelf-life and ambient temperature storage, compared to liquid reaction mixes, by using wax layers to isolate enzymes from other reagents; (2) improved specificity compared to other LAMP end-point reporting methods, by using sequence-specific QUASR (quenching of unincorporated amplification signal reporters); (3) increased sensitivity, compared to methods without purification through use of a magnetic wand to enable pipette-free concentration of sample RNA and cell debris removal; (4) quality control with a nasopharyngeal-specific mRNA target; and (5) co-detection of other respiratory viruses, such as influenza B, by multiplexing QUASR-modified RT-LAMP primer sets. The flexible nature of the ALERT workflow allows easy, at-home and point-of-care testing for individuals and higher-throughput processing for labs and hospitals. With minimal effort, severe acute respiratory syndrome coronavirus 2 (SARS-CoV-2)-specific primer sets can be swapped out for other targets to repurpose ALERT to detect other viruses, microorganisms, or nucleic acid-based markers.

## 1. Introduction

The coronavirus disease 2019 (COVID-19) pandemic has brought the field of molecular diagnostics into the spotlight. Since the initial release of sequence data for severe acute respiratory syndrome coronavirus 2 (SARS-CoV-2) on 10 January 2020 (GenBank Accession # MN908947), a positive-sense single-stranded RNA of approximately 29.8 kb, reverse transcription-polymerase chain reaction (RT-PCR) assays have been designed by the World Health Organization (WHO), the US Centers for Disease Control (CDC), the Chinese CDC, Institut Pasteur and others [1]. An unprecedented demand for RT-PCR diagnostics has put a strain on every aspect of conducting these laboratory-based assays. Despite the availability of quickly developed protocols (e.g., [2,3,4,5]), shortages in materials (e.g., swabs, reagents, and consumables) and of infrastructure (e.g., approved facilities, technicians, and equipment) have prevented the efficient testing, tracing, and isolation of infectious individuals. These shortages have led to a global public health crisis of unforeseen consequences, which is further exacerbated in the global south.

Conventional RT-PCR-based assays, which require a level of training, are neither rapid, inexpensive, nor highly scalable without the use of cost-prohibitive equipment. To address the challenge of diagnostics at scale there has been an increased emphasis on simpler detection techniques conducted at the point-of-care or even in the convenience of one’s own home. Most of these methods can be broadly categorized into two groups in relation to their targets, nucleic acids or proteins (i.e., antigen). Amongst the rapid methods targeting nucleic acids, reverse transcription loop-mediated isothermal amplification (RT-LAMP) [6] has risen in prominence during the course of the COVID-19 pandemic. Beyond its isothermal nature, which eliminates the need for complex instruments such as thermocyclers necessary for PCR, RT-LAMP is attractive since it produces an immense amount of amplification products allowing visual detection by the naked eye via diverse methods [7]. Since the emergence of COVID-19, multiple primer sets for SARS-CoV-2 [8,9] and variations on the RT-LAMP method have been published [10] and some have received emergency use authorization by the Food and Drug Administration of the USA (Color and Mammoth Biosciences). While the RT-LAMP workflow delivers results faster than RT-PCR and is highly scalable, it still requires a trained technician to perform the assay as well as cold shipping and storage of reagents. Accessible LAMP-Enabled Rapid Test (ALERT) was developed to address some of the shortcomings. A simple workflow integrates low-melting point waxes for the stable packaging of reaction components together with a magnetic-wand based technique for the pipette-free movement of concentrated RNA from sample to reaction vessel (Figure 1).

## 2. Materials and Methods

### 2.1. Controls and Reference Material

SARS-CoV-2 synthetic RNA controls were purchased from Twist Biosciences (San Francisco, CA, USA, SKU: 102019) and influenza B virus RNA from ATCC (Catalogue Number VR-1813D).

SARS-CoV-2 virus was cultured at the Innovative Genome Institute’s BSL3 facility at University of California Berkeley and inactivated in RNAShield at a 2.5 × 10^5^ PFU/mL concentration. Samples were spiked with virus at units ranging between 125 PFU and 2000 PFU per RT-LAMP reaction. These samples were processed in BSL2-dedicated rooms following strict biosafety guidelines.

RNA positive controls for the reactions performed in Chile were created by in vitro transcription using the Hi-Scribe kit from NEB (catalogue E2040S) from an amplicon obtained by PCR amplification of the IDT control for Sars-CoV-2 (catalogue 10006625) using the following primers: NT7_Fw (CGA AAT TAA TAC GAC TCA CTA TAG GGG CAA CGC GAT GAC GAT GGA TAG) and T7_Nter_Rv (ACT GAT CAA AAA ACC CCT CAA GAC CCG TTT AGA GGC CCC AAG GGG TTA TGC TAG TTA GGC CTG AGT TGA GTC AG).

In vitro transcribed RNA was treated with DNAse I (M0303S) for 15 min at 37 °C before purification with Qiagen RNeasy kit, and serially diluted to the concentrations used.

### 2.2. Sample Collection

Human nasal samples were collected with either nasopharyngeal (NP) swabs, nasal mid-turbinate (NMT) swabs, or a nasopharynx flush-through (NFT) method. NFT samples were collected by using a 5 mL dropper filled with water squirted into a nostril with the head tilted back and the water passing through the nasopharynx and subsequently spit into a 50 mL Falcon tube.

For collecting nasal mid-turbinate samples, a variety of swabs was used due to supply chain shortages, ranging from Q-tip cotton swabs to flocked nylon nasopharyngeal swabs. It was observed that some sample collection swabs, such as those supplied by Zymo Research Corporation (C105250), had blue fluorescence emission when excited by ultraviolet (UV) light thus interfering with the interpretation of results following crude sample lysis protocols. Ultimately sterile, flocked oral swabs with a 30 mm breakpoint compatible with 1.5 mL microcentrifuge tubes (SNT Biotech, Plainfield, IL, USA, www.sntbiotech.com, accessed on 2 January 2021) was chosen for sample collection.

### 2.3. Sample Lysis

Different lysis buffers were experimented with. The proteinase K (PK) extraction method begins by inserting a freshly used NP or NMT swab into 300 µL of PK Lysis Solution and agitating the swab for 30 s. 75 µL of the resulting solution is used and, depending on the experiment, spiked with inactivated SARS-CoV-2. Samples were heated at 55 °C for 5 min for lysis and 95 °C at 10 min for inactivation. The PK Lysis Solution contained ~15 mAU/mL and a 10 mL stock was prepared by diluting 0.25 mL proteinase K (Qiagen, Hilden, Germany, 19131) with 9.75 mL water.

In order to process samples already in solution, such as PBS, nasal swabs (in an unknown solution), and saliva sent by the XPRIZE foundation (www.xprize.org/prizes/covidtesting, accessed on 10 October 2020) QuickExtract DNA extraction solution was used (LGC-Lucigen, Madison, WI, USA, catalogue # QE09050) following the protocol by Kellner et al. (2020).

For a do-it-yourself (DIY) workflow, the buffer of choice was TCEP/EDTA following the protocol outlined by Rabe and Cepko (2020) of a modified HUDSON buffer (Myhrvold et al. 2018). In contrast to other protocols where a concentrated 100X HUDSON buffer is added to a liquid sample such as saliva, a 1× solution (2.5 mM TCEP, 1 mM EDTA) was used for the direct addition of an NMT swab.

### 2.4. Nucleic Acid Extraction

RNA from NP, NMT, and NFT samples was isolated using both a QIAamp Viral RNA Mini Kit (Qiagen Catalogue # 52904) according to the manufacturer’s instructions as well as with a TRIzol/chloroform method [11]. NP and NMT swabs were inserted into 1ml of phosphate-buffered saline (PBS) and vortexed prior to extraction while 1 mL of NFT samples was used directly for extraction.

### 2.5. RNA Isolation with Paramagnetic Beads and Magnetic Wand

For paramagnetic bead-based RNA capture, 0.6 volumes of a 1× bead solution was added to the lysed sample and uniformly suspended. After a 5 min incubation at room temperature, the samples containing the beads were placed on an appropriately sized magnetic rack and beads were pelleted by the magnet for 5 min. All liquid was removed, and a brief wash was conducted with 100 µL of 85% ethanol. Following the removal of ethanol, sample tubes were removed from the magnetic rack and the beads were resuspended in 10–20 µL of water. Depending on the cartridge design, 5–10 µL of beads in water were directly added to the cartridge. The 1× bead solution is prepared by combining one part RNAClean XP beads (Beckman Coulter Life Sciences, Indianapolis, IN, USA, A63987) with five parts of a concentrated Bead Binding Buffer. The composition of the Bead Binding Buffer [12] is 20% PEG 8000 (Sigma-Aldrich, Munich, Germany, P1458), 10 mM Tris HCl, pH 8.0 (Invitrogen, Waltham, MA, USA, 15568025), 0.05% Tween 20 (Thermo Scientific, Waltham, MA, USA, J20605AP), 5 mM sodium azide (MP Biomedicals, Irvine, CA, USA, 0210289125), sodium chloride (Fisher Chemical, Hampton, NH, USA, S271-500).

To enable a DIY workflow, 3D printed the sheaths were used for the flanged magnetic wand with a $450 3D printer (Adventurer 3; Flashforge) in as few as 10 min using about $0.02 worth of 1.75 mm diameter PLA filament (Flashforge) per wand. These flanged sheaths accommodate a $0.72 cylindrical grade N42 neodymium magnet (1-inch-long × 1/16-inch diameter; D1 × 0, K&J Magnetics, Inc., Jamison, PA, USA). The diameter and vertical position of the flange can be adjusted to accommodate a diverse array of sample tubes. The diameter of the rest of the sheath is optimal for depositing the beads in RT-LAMP reaction cartridges that have been prepared in 0.2 mL strip tubes. If larger tubes are used for the RT-LAMP reaction cartridge, the diameter of the sheath can be increased to accommodate more powerful, larger diameter magnets. Using these designs as a prototype, the magnetic wand sheaths can also be produced in bulk with injection-molded low-retention plastics.

Capturing the beads with magnetic wands followed a similar protocol to using a magnetic rack but without the use of pipettes. After the 5-min room temperature incubation the magnetic wand was inserted into the sample/bead solution. A further 5-min room temperature incubation was sufficient to allow RNA-bound beads to be attracted by the wand. The wand, now with beads, was removed from the sample. At this stage an optional bead wash can be performed by briefly dipping (~2 s) the magnetic wand into a tube with 85% ethanol filled to the same volume as the sample/bead solution. If an ethanol wash was performed, beads on the magnetic wand were allowed to air dry without over-drying beads (~2–5 min). The magnetic wand was placed in a tube containing eluent (in this case, RT-LAMP cartridges with 10 µL water on the topmost layer of wax). The magnet was removed from the wand and the wand was swirled to release beads into the eluent.

### 2.6. Construction of Wax-Layered Cartridges

For a sampling and process control target sequence, the BPI fold containing family A, member 1 gene (BPIFA1) was chosen as its expression is nasopharyngeal specific [13]. A LAMP primer set for this gene was designed that traverses an exon-exon boundary, making the reaction specific to BPIFA1 mRNA. The SARS-CoV-2 [8,10] and influenza B [14] primers were adopted from the literature but modified for the sequence specific QUASR (quenching of unincorporated amplification signal reporters) reporting system [15] (Appendix A).

#### 2.6.1. One-Step Reverse Transcription Loop-Mediated Isothermal Amplification (RT-LAMP)

One-step RT-LAMP cartridges using the NA/NB primer set were prepared by adding 27 µL of RT-LAMP primer mix to the bottom of a 0.75 mL cryogenic tube (Micronic, Lelystad, The Netherlands) followed by 200 µL of melted silicone wax (Siltech, Toronto, ON, USA, D-222, melting point 37 °C). After solidification of this wax layer a small divot was made with a 1 mL micropipette tip, deposited into which was 3 µL of an enzyme mix consisting of 1 µL (15 units) of WarmStart RTx (NEB, Ipswich, MA, USA, M0380) and 2 µL (16 units) of Bst 3.0 (NEB, Ipswich, MA, USA, M0374). The enzyme droplet was capped by a further addition of 300 µL of the same silicone wax. In an alternative construction of this cartridge, a layer of 150 µL cold mineral oil (consumer grade) was added between the enzyme droplet and silicone wax to shield it from the thermal shock of hot wax.

For the single wax layer RT-LAMP cartridges using this NA/NB primer set, the primer mix, prepared in isothermal amplification buffer (I) (NEB, Ipswich, MA, B0537S) contained 2.1 µM each of NB-FIP and NB-BIP, 0.32 µM each of NB-F3 and NB-B3, 1.3 µM each of NB-LB and NB-LF-Tx and 2.2 µM of NB-LF-Q. MgSO_4_ concentration was at 7.8 mM and dNTPs at 1.8 mM. 27 µL of the primer mix was used as the bottom layer of reaction cartridges anticipating a 35 µL final reaction volume together with 1 µL WarmStart Rtx (NEB, Ipswich, MA, USA, M0380), 2 µL Bst 3.0 (NEB, Ipswich, MA, USA, M0374) and 5 µL of sample.

For direct addition of paramagnetic beads using the magnetic wand, the NM primer set (Appendix A) was used in a 30 µL final reaction volume cartridge consisting of (from bottom to top) a 16.5 µL primer mix, a layer of paraffin wax (IGI 1250A melting point of 61.4 °C), and an enzyme mix of 1.5 µL (22.5 units) of WarmStart RTx (NEB, Ipswich, MA, USA, M0380) and 2 µL (16 units) of Bst 2.0 (NEB, Ipswich, MA, USA, M0374). The enzyme was trapped between two layers of solid wax by adding a final 4 mm solid, lower-melting-point silicone wax (Siltech, Silwax D-222, melting point of 37 °C) and melting it down briefly at 40 °C for 1 min. When the magnetic wand was used, 10 µL of molecular biology-grade water for bead deposition was added above this wax layer. (Figure 2).

For one-step cartridges using the NM primer set, the 16.5 µL primer mix was prepared with isothermal amplification buffer (I) (NEB Ipswich, MA, B0537S) with 2.9 µM NM-FIP-F and NM-BIP, 4.35 µM NM-FIP-Q, 1.5 µM NM-LF and NM-LB, and 0.4 µM NM-F3 and NM-B3. MgSO_4_ was at 12.8 mM and dNTPs at 2.5 mM.

Depending on the primer mixes, 5 µL (for NA/NB) or 10 µL (for NM) of sample was added to the top layer. Cartridges containing the NA/NB set were incubated in a 63 °C dry block with a heated lid for 1 h. Cartridges containing the NM primer set, and where the enzyme was packaged between waxes of two different melting points, tubes were first heated to 64 °C to allow the mixing of all reagents and then cooled to 55 °C for a 12.5 min reverse transcription reaction, followed by a 64 °C LAMP incubation for 45 min to 80 min.

For the BPIFA1 primer set, standard RT-LAMP primer concentrations were followed: 1.6 µM FIP and BIP, 0.4 µM LF and LB-F, 0.2 µM F3 and B3, and 0.6 µM LB-Q. 7 mM MgSO_4_ and 1.4 mM dNTPs were used in a 25 µL reaction in 1× Isothermal Amplification Buffer (I) (NEB Ipswich, MA, USA, B0537S) with 0.75 µL WarmStart Rtx (NEB, Ipswich, MA, USA, M0380) and 1.5 µL Bst 2.0 (NEB, Ipswich, MA, USA, M0538L). BPIFA1 RT-LAMP reactions were incubated at 63 °C for 30 min.

A wax-layered cartridge that is ready to run following sample addition presents a challenge towards the homogenous mixing of all of its components owing to a pipette-free operation. It was found that the addition of a 2 mm glass bead to the top of the tube allows for efficient and thorough mixing by both allowing for reagents to fall through wax layers as well as displacing the bottommost primer mix.

#### 2.6.2. Two-Step RT/LAMP

Cartridges separating the reverse transcription (RT) step from the loop-mediated isothermal amplification step (LAMP) were constructed into 8-tube strips using two separate waxes with distinct melting points. The 10 µL LAMP primer mix, at the bottom of the cartridge, was sealed from the RT primer mix above it with a layer of paraffin wax (IGI 1250A; melting point of 61.4 °C). This seal was created by adding a roughly 3 mm diameter piece of solid wax above the LAMP primer mix and melting it at 65 °C for 1 min. After solidification of this wax layer at room temperature, 7 µL of RT primer mix was added above it. Two drops of a lower melting point silicone wax (Siltech, Silwax D-222; melting point of 37 °C), heated to 50 °C in a water bath, was added on top of the RT primer mix using a 1 mL micropipette and allowed to solidify at room temperature. Finally, an enzyme mix consisting of 1 µL (15 units) WarmStart RTx (NEB, Ipswich, MA, USA, M0380) and 2 µL (16 units) of Bst 3.0 (NEB, Ipswich, MA, USA, M0374) was added as the top layer of the cartridge (Figure 3).

For the 2-step detection cartridges, two separate primer mixes corresponding to the reverse transcription (RT Mix) and loop-mediated isothermal amplification (LAMP Mix) steps were made.

The RT Mix contained the F3 outer primer from the N-A primer set [8] together with the B3 primer from the N-B primer set, which was found to improve the detection limit compared to using only the NB-B3 primer. This may result from DNA polymerase activity stemming from the inclusion of Bst 3.0 during the RT step. NA-F3 and NB-B3 were at 0.54 mM along with 12.9 mM of MgSO4 and 1.07 mM of dNTPs in NEB isothermal amplification buffer (I) at a 2× concentration. In each cartridge 7 µL of this mix was used in anticipation of a 15 µL reverse transcription reaction together with 1 µL WarmStart Rtx (NEB, Ipswich, MA, USA, M0380), 2 µL Bst 3.0 (NEB, Ipswich, MA, USA, M0374), and 5 µL of sample.

The LAMP Mix was composed of the NB primer set published by Zhang et al. (2020) but modified for the sequence-specific QUASR reporting system. This mix, with the NEB isothermal amplification buffer (I) (NEB Ipswich, MA, B0537S) at a 1× concentration, had primer concentrations of 4 µM NB-FIP and NB-BIP, 0.625 mM NB-F3 and N-B-B3, 2.5 µM NB-LB and NB-LF-Tx, and 5 µM of NB-LF-Q. MgSO_4_ concentration was at 6 mM and dNTPs at 3.5 mM. The concentration of the LAMP mix is higher than standard reactions; the final LAMP reaction volume will be 25 µL when all reagent mixes are combined at the bottom of the tube. In each cartridge 10 µL of this LAMP mix was used.

The 2-step cartridges had 5 µL of sample added to the enzyme droplet on the top layer of lower melting point silicone wax, incubated in a thermal cycler at 55 °C for 2 min allowing for the enzyme droplet and sample to fall through liquid wax and meet the RT primer mix. Cartridges were removed from incubation and the wax was allowed to briefly solidify in order to allow for mixing of reagents by manual shaking. After mixing, cartridges were further incubated for 20 min at 55 °C for the RT step to take place (Figure 2b). After this step, the temperature was raised to 63 °C for 2 min to allow the RT reaction to fall through the now liquefied paraffin wax layer and meet the LAMP primer mix. The cartridges were again removed from incubation, wax layers allowed to solidify, and the RT-LAMP mixtures, now together under a layer of solid silicone and another of paraffin wax, mixed briefly by vortex. The LAMP reaction was then allowed to progress for 45 min at 63 °C.

### 2.7. Multiplex RT-LAMP Reactions

The final 25 µL reactions, prepared with 1X Isothermal Amplification Buffer (I) (NEB Ipswich, MA, USA, B0537S) had 1.6 mM of both FIP and BIP primer pairs, 0.2 mM of influenza B or BPIFA1 F3 and B3 primer pairs, 0.25 µM of the NB-F3 and NB-B3 primer pair, 1 µM of both LF and LB primer pairs (LF or LB being modified with a fluorophore) and 2 mM of the corresponding LF or LB quencher primer, 6 mM MgSO4, 1.6 mM dNTPs, 1 µL (15 units) of WarmStart RTx (NEB, Ipswich, MA, M0380), and 2 µL (16 units) of Bst 3.0 (NEB, Ipswich, MA, USA, M0374). Reactions were incubated at 63 °C for 45 min.

### 2.8. Visualization

End-point reactions were visualized on an ultraviolet (UV) transilluminator. Alternatively, for reactions utilizing fluorescein tagged oligonucleotides the low-cost (<$2) open-source GMO Detective fluorescence detector, with 470nm LEDs and orange and blue gel filters was used [16] (https://github.com/MakerLabCRI/GMODetective-Detector, accessed on 11 December 2020).

### 2.9. Incubation

Incubations (55 °C and 63 °C or 64 °C) were performed in a thermocycler. All reactions were also successfully performed in a dry block incubator with a heated lid (Benchmark Scientific, Sayreville, NJ, USA, BSH200-HL).

### 2.10. Analytical Validation for 2-Step RT-LAMP: XPRIZE Blinded Proficiency Test

As a semi-finalist in the XPRIZE Covid Testing challenge [17] we had the opportunity to participate in a blinded proficiency test alongside 218 other teams. Two separate 96-well sample plates were mailed to the Oakland Genomics Center in Oakland, CA, with one consisting of 86 synthetic RNA samples spiked into water (SARS-CoV-2 and 15 other viruses to test cross-reactivity) shipped on dry ice, and the other plate consisting of 67 samples of different matrices (phosphate buffered saline (PBS), nasal swabs resuspended in an unknown buffer, and saliva) spiked with SARS-CoV-2 particles from Zeptometrix shipped with cold packs. The synthetic RNA plate was kept at −80 °C and the Zeptometrix plate was kept at 4 °C until analysis. For RNA extraction of the XPRIZE Rapid Covid Testing test samples, 200 µL of QuickExtract DNA Extraction Solution (Lucigen, QE09050) was added to 200 µL of each sample. Samples were incubated for 5 min at room temperature prior to inactivation of the QuickExtract solution at 95 °C for 10 min. The samples were cooled on ice to room temperature prior to adding 240 µL of bead solution as described above. After thoroughly mixing the beads into the samples, beads were allowed to bind RNA for 5 min at room temperature prior to placing the samples on a magnetic rack for 5 min to pellet the RNA-bound beads. The supernatant was removed from each sample and each set of remaining beads was washed for 5 min with 850 µL 85% ethanol. After removal of ethanol, the beads were allowed to air dry briefly. RNA was eluted from beads with 10 µL of nuclease-free water and added to the 2-step RT-LAMP cartridges as described above.

### 2.11. Clinical Validation for 2-Step RT-LAMP

In order to validate detection cartridges with true clinical samples from patients diagnosed for the presence of absence of SARS-CoV-2 using established RT-PCR methods v1 2-step RT-LAMP cartridges were shipped, at ambient temperature, to locations willing to share clinical samples. At the Hôpital Saint Louis (Paris, France), 18 samples, determined to be positive with RT-PCR methods approved by local agencies, (Ct values ranging between 13 and 37.9) were analyzed alongside a negative water control and Coronavirus NL63 for cross-reactivity. At the Pontificia Universidad Católica de Chile (Santiago, Chile), 10 negative patient samples and 19 samples, determined to be positive with RT-PCR using the N1 and N2 primer set designed by the CDC, (Ct values ranging between 20.25 and 35.34) were analyzed. These clinical samples were obtained from anonymous patients that attended the outpatient service of Red Salud UC-CHRISTUS (Santiago, Chile). All methods were performed in accordance with the relevant guidelines and regulations. All procedures were approved by the Ethics Committee of the Pontificia Universidad Católica de Chile.

## 3. Results

We have developed a streamlined process for detecting nucleic acids of RNA viruses, demonstrated as a proof-of-principle here for the detection of SARS-CoV-2 from nasal mid-turbinate swabs. The ALERT system incorporates a room temperature stable lysis buffer (a modified HUDSON buffer or proteinase K) for the lysis of virions as well inactivation of RNAses, followed by bead-based sample concentration in which the beads are captured with a magnetic wand developed for pipette-free testing. The RT-LAMP reagents are packaged into a reaction vessel constructed with layers of waxes in-order to increase shelf-life at room temperature, provide physical stability for a just-add-sample operation as well as to minimize risk of cross-contamination.

A substantial improvement in the limit of detection (LoD) of the first RT-LAMP primer set published (Zhang et al., 2020) for SARS-CoV-2 was achieved by separating the RT reaction from the LAMP reaction using waxes of different temperatures (2-step cartridges). To facilitate simpler production and usage of cartridges a 1-step design using another published primer set [10] with improved LoD was ultimately preferred.

### 3.1. Lysis

Proteinase K was used to lyse NMT swabs from 11 distinct individuals (asymptomatic and presumed to be uninfected with SARS-CoV-2) spiked with inactivated SARS-CoV-2 ranging from 250 to 1000 PFU. Lysed samples were concentrated using RNAClean XP paramagnetic beads and SARS-CoV-2 RNA detected using a master mix containing the NM primer set. All spiked samples presented fluorescence associated with amplification while non-spiked samples did not (Figure 4a).

In order to reduce cost and eliminate the 55 °C incubation step of Proteinase K a modified HUDSON buffer made of TCEP and EDTA was used [18]. NMT swabs were deposited into tubes with 300 µL 1X HUDSON buffer and spiked with inactivated SARS-CoV-2 at 0, 125, 250 and 750 PFU. After a brief vortex, samples were incubated for 5 min at 95 °C. These samples were concentrated with paramagnetic beads and 10 µL of a 20 µL suspension of beads in water was deposited into a one-step cartridge containing the NM primer set. All spiked samples presented fluorescence associated with amplification while non-spiked samples did not (Figure 4b).

### 3.2. Magnetic Wand Sample Deposition

In order to produce a streamlined extraction method compatible with an at-home test a 3D-printed magnetic wand was used to directly deposit paramagnetic beads into RT-LAMP reaction cartridges. In its simplest configuration, the magnetic wand consists of a cylindrical neodymium rare-earth magnet that slides into a 3D-printed sheath closed on one side. A typical bead pulldown using a conventional magnetic rack or plate incorporates a 5 min incubation with the paramagnetic beads. To make the magnetic wand more user-friendly with such incubations, a version of the magnetic wand that incorporates a flange that rests on the rim of the sample tube and allows the tip of the wand to be optimally positioned in the sample was designed.

To demonstrate the magnetic wand, RNA-bound paramagnetic beads were deposited into a one-step reaction cartridge with 10 µL of molecular biology grade water at the top to elute the beads into. NMT swabs in proteinase K were spiked with 125 to 1000 PFU of inactivated SARS-CoV-2 in duplicate and processed according to the proteinase K extraction and bead purification protocol. Paramagnetic beads were recovered with the magnetic wands and directly deposited into the 10 µL of water after a brief dip in 100 µL 85% ethanol. All concentrations of virus down to 250 PFU gave a bright signal while one of the two repeats of 125 PFU appeared to amplify (Figure 5a). In addition to visualizing these end-point reactions using a UV transilluminator, GMO Detective [16], an approximately $2, open-source, fluorescence visualizer to observe the results was used (Figure 5b).

### 3.3. Improvement of Limit of Detection (LoD) in a 2-Step Reaction

The N-B (and N-A) set published by Zhang et al. (2020) was reported to have a limit of detection around 480 copies of synthetic SARS-CoV-2 RNA per 25 µL reaction and as confirmed here. Separating the 1-step RT-LAMP reaction into a 2-step RT-LAMP reaction decreased the LoD to 25 copies of synthetic SARS-CoV-2 RNA (Figure 6).

### 3.4. Primer Specificity for BPIFA1 mRNA

In order to test that the BPIFA1 RT-LAMP set, designed to traverse an exon-exon junction (Figure 7a), is indeed specific to nasopharyngeal mRNA a NEB WarmStart Colorimetric RT-LAMP kit was used (NEB, Ipswich, MA, USA, M1800) according to the manufacturer’s instructions. A variety of samples including saliva and blood RNA as well as human genomic DNA were used to screen this primer set (Figure 7b). The robustness of the BPIF1A primer set was demonstrated by running it alongside spiked nasal samples from 11 individuals (Figure 7c).

### 3.5. Multiplexing: Co-Detection of Severe Acute Respiratory Syndrome Coronavirus 2 (SARS-CoV-2) with Influenza B or Endogenous Control Gene BPIFA1

In order to demonstrate the feasibility of multiplex detection of a nasopharyngeal endogenous control mRNA sequence BPIFA1 and SARS-CoV-2, extracted human RNA was spiked with synthetic SARS-CoV-2 RNA (4000 copies) and a clear tricolor result was observed depending on target input. This type of tissue-specific target acts both as a control for proper sample collection as well as an internal control for the RT-LAMP reaction (Figure 8a).

NFT samples were collected from 8 distinct individuals (asymptomatic and presumed to not be infected with SARS-CoV-2). Each sample was split into two aliquots, one of which was spiked with 500 PFU of inactivated SARS-CoV-2. Spiked and non-spiked samples were extracted with the TRIzol/Chloroform protocol. Every individual and their corresponding spiked sample gave a clear binary signal of green (BPIFA1 amplification) vs. red (SARS-Cov-2 amplification) or orange (BPIFA1 and SARS-CoV-2 amplification). A reduction of LoD was noticed with the multiplex primer mix as the previous reliable detection of 25–50 copies of synthetic SARS-CoV-2 RNA was not achieved (Figure 8b). Further optimization is needed to reliably co-detect endogenous control target BPIFA1 and SARS-CoV-2.

To demonstrate the possibility of detecting multiple RNA viruses in a single reaction an influenza B RT-LAMP primer set was multiplexed together with the SARS-CoV-2 specific N-B set showing a clear tricolor result depending on sample input (Figure 8c).

### 3.6. Analytical Validation: XPRIZE Blinded Proficiency Test

The format for the XPRIZE Rapid Covid Testing semi-finalist round enabled validation and evaluation of multiple parameters of the assay, including sensitivity, specificity, functionality in different sample matrices, and cross-reactivity. Viral material in the samples was present either as non-infectious, purified, intact SARS-CoV-2 viral particles (NATSARS(COV2)-ERC; ZeptoMetrix Corporation, Buffalo, NY, USA) or synthetic RNAs representing the full genome of SARS-CoV-2 or other respiratory viruses (Twist Bioscience, San Francisco, CA, USA).

Sensitivity and specificity were evaluated using samples with different concentrations of SARS-CoV-2 ZeptoMetrix particles (0.1, 0.5, 1, 2, 5, 10, 25, 50, 100 copies/µL), Twist synthetic RNAs (0.01, 0.02, 0.05, 0.1, 0.5, 1, 2, 3, 4, 5, 10, 25, 50, 100, 200, 500, 1000, 10,000 copies/µL), or mock negative samples. Given the variability in quality of biological samples and the presence of contaminants and foreign material that can potentially interfere with the downstream assay, SARS-CoV-2 ZeptoMetrix particles were spiked into the three different matrices: nasal samples in an unknown solution, saliva samples and 1× Phosphate-buffered saline (1× PBS). Since Twist synthetic RNAs are neither protected within a host cell nor a viral envelope, they were spiked into water. Cross-reactivity was evaluated using synthetic controls from Twist Bioscience for a wide range of respiratory viruses, including coronaviruses (SARS-CoV-1, MERS, 229E, NL63, OC43), influenza viruses (influenza A/H1N1, influenza A/H3N2, influenza B), enteroviruses (D68, HRV strain 89), paramyxoviruses (measles, mumps, parainfluenza 1, parainfluenza 4), and human bocavirus 1.

The blinded test set comprised 67 samples with an unknown concentration of SARS-CoV-2 ZeptoMetrix particles (5 to 7 replicates per concentration), 56 samples with an unknown concentration of SARS-CoV-2 Twist synthetic RNAs (1 to 5 replicates per concentration), and 30 samples with 500 copies/µL of fifteen non-SARS-CoV-2 Twist synthetic RNAs in duplicate. In addition to the blinded test samples, a pair of known positive and negative controls was included for both the ZeptoMetrix particles (100 copies/µL and 0 copies/µL, respectively) and Twist synthetic RNAs (10,000 copies/µL and 0 copies/µL, respectively). Upon receiving the XPRIZE Rapid Covid Testing samples, competing teams were informed only of the matrix of each sample and which four samples were the positive and negative controls.

Virus concentrations as low as 0.1 copies/µL and RNA as low as 2 copies/µL were successfully detected (Appendix A). The LoD (with a 95% true positive rate) with the blinded XPRIZE samples was in the range of 0.1 to 2 copies/µL for the virus-containing samples and 25 to 50 copies/µL for the synthetic RNA-containing samples (Table 1, Figure 9). None of the six mock negative samples was scored as a positive result. One of the two cross-reactivity controls for SARS-CoV-1 had moderate fluorescence and was scored as a positive result. Assays for all of the other cross-reactivity controls had the expected negative result. This suggests a true negative rate of 97% to 100%, depending on whether cross-reactivity test samples are considered in the calculation (Table 2). Of the 219 XPRIZE Rapid Covid Testing semi-finalists, ALERT performed well enough to advance the ViralAlert team to be one of 20 finalists in the competition.

### 3.7. Clinical Validation and Stability in Shipping

To test the v1 NA/NB two-step cartridge system with patient samples, cartridges were prepared at the Oakland Genomics Center (Oakland, CA, USA) and shipped at ambient temperature conditions to two different sites with access to RNA extracted from NP swabs from patients. Cartridges utilized at the Hôpital Saint-Louis (Paris, France) were kept at ambient temperature for 16 days from preparation until use while cartridges ran at Pontificia Universidad Católica de Chile (Santiago, Chile) were kept at 4 °C following a 4-day shipping at ambient temperature.

At the Hôpital Saint-Louis, the positive predictive value was found to be 84% in comparison to the in-house RT-PCR method and no cross reactivity with coronavirus NL63. Unfortunately, at this site no negative patient samples were examined. A single non-template water control was run alongside the samples which did not produce an amplification signal (Appendix A).

At the Pontificia Universidad Católica de Chile a more detailed analysis of results was undertaken after some samples were observed to have faint fluorescence. Images of tubes were captured using both a high-exposure and a low-exposure setting on a digital camera. In the high-exposure setting, fluorescence was observed in all true positive samples but the negative predictive value of the 2-step NA/NB cartridge was found to be at 60%. At low exposure, there were no false positive samples while the positive predictive value fell to 74% in addition to false negative RNA controls (Appendix A, Appendix A).

In the past we have constructed similar wax layered detection cartridges with accurate results after 1 month of storage at room temperature. The enzymes which catalyze the RT-LAMP reaction are stable for up to 3 months in the glycerol suspension they are provided in (personal communication, New England Biolabs) and, therefore, the cartridge design separating the enzymes from the reaction buffer should provide comparable stability at room temperature. These shelf-life experiments are currently underway.

## 4. Discussion

The applicability of RT-LAMP for the detection of SARS-CoV-2 has been shown to be robust by numerous authors [19,20], undergone clinical validation [21], and has been granted Emergency Use Authorization by the FDA (Color Health, Burlingame, CA, USA; Atilla Biosciences, Mountain View, CA, USA and Mammoth Biosciences, Brisbane, CA, USA). As noted by Kellner et al. (2020), “There are, however, several remaining challenges, especially with respect to the world-wide distribution and access to RT-LAMP reagents. All of our assays were performed with commercial reagents that require −20 °C storage, which is prohibitive for low-resource settings. Also, pre-aliquoted ready-to-use reaction mixes required for home-testing may be less stable even when stored at low temperature.” The Accessible LAMP-Enabled Rapid Test (ALERT) system packages the components of the well-established RT-LAMP reaction amongst layers of low-melting point waxes in order to provide room temperature and physical stability resulting in an easy to use “just-add-sample” solution for molecular diagnostics. Lyophilization is one solution to ensure stable reagents and has been applied to RT-LAMP reagent mixes for SARS-CoV-2 [22,23]. Others have also shown that simple drying of reagents is sufficient [24]. Here it is demonstrated that liquid reagents packaged tightly amidst wax layers are shielded from physical and environmental disturbances and are ready to run immediately upon receipt. Although freeze-drying and drying methods are currently more suitable for high-throughput manufacturing, the waxes employed in the ALERT system are available in mm scale solid pellets, potentially allowing for automatic deposition followed by a brief heating step to melt the wax, and seal reagents.

RT-LAMP cartridges utilizing wax layering present several advantages over conventional one-step RT-LAMP reactions. As demonstrated, waxes can be harnessed for the separation of the reverse transcription reaction from the loop-mediated isothermal amplification reaction which can increase the sensitivity of an assay 10-fold. The wax layers also allow for physical stability of the reaction cartridge since all the reagents are kept tightly in place and the separation of the enzymes from the reaction buffer increases the shelf-life of cartridges at room temperature. Before enzymes with aptamer modifications were introduced into the market paraffin waxes were used as a hot-start method and the system presented here also allows for the reaction to begin at the optimal annealing temperature for the LAMP primers. Further, similar to the mineral oil layer used prior to heated lids of thermocyclers, a wax layer above the reaction prevents evaporation and volume loss. The preferred lysis solutions, proteinase K and TCEP/EDTA, were also chosen for their stability at room temperature.

In molecular diagnostics of pathogens, the sample collection, extraction, and purification steps are as imperative as the final step of molecular detection. While it is important to have a sensitive test, such a test is useless without the refinement of these upstream processes.

Paramagnetic beads with special coatings or adaptations have been used for decades to isolate, purify, and/or concentrate nucleic acids [25]. In recent years, use of such beads has become ubiquitous in labs that synthesize DNA/RNA libraries for next-generation sequencing. The nucleic acid-bound beads are typically concentrated and removed from solution with the use of magnetic racks or plates that can accommodate a variety of tube formats from 96-well plates to 50 mL conical tubes. These robust and long-lasting magnetic devices are well-suited for a lab setting. With a typical cost from $300–$1000 per device, such specialized equipment can be cost-prohibitive. Together with the additional requirement of pipettes, they can also be too fragile or bulky for use in the field and require some degree of training to use. To avoid these issues, the “magnetic wand” was designed as an alternative. Use of the <$1 magnetic wand with RNA-binding paramagnetic beads results in a higher final concentration of the biological materials actually tested, thereby greatly increasing the sensitivity of the assay.

Paramagnetic bead-based extraction methods allow for the concentration of RNA from a large sample volume and are, therefore, amenable to sample pooling strategies [12,26]. Additionally, paramagnetic beads can be modified to capture a specific sequence of RNA and, therefore, even further increasing the sensitivity of the molecular diagnostic pipeline. With the magnetic wand method of moving beads from extraction into detection an inexpensive, and easy-to-use alternative to pipettes and magnetic racks or plates is introduced to readily empower at-home and point-of-care users to perform testing. Magnetic wands can also be configured for high-throughput processing of many samples at once.

Robust molecular diagnostics require carefully chosen controls. The most common targets chosen as endogenous controls for RT-PCR and RT-LAMP reactions are housekeeping genes such as RNAse P [27] and GAPDH [28] which do not have tissue-specific expression patterns. Recently, an RT-LAMP primer set has been designed to detect the saliva-specific expression of STATH [29] to work in conjunction with saliva-based detection of SARS-CoV-2. This primer set adds to an increasing number of tissue-specific mRNA RT-LAMP primer sets by designing one specific to nasopharynx mRNA.

An essential consideration regarding separation of reaction components by layers of wax is the ultimate, homogeneous mixing of the primer mix, enzymes and sample. While convective currents during incubation and manual agitation help in this regard, they are not as thorough as mixing with micropipettes. We suspect that the few false negative results observed during clinical testing in France and Chile that were not explained by high Ct values from RT-PCRs on the same samples, may be associated with the heterogeneity of the reaction mix. A number of solutions can be iterated upon the ALERT system such as an addition of solid glass beads to the reaction for increased physical mixing or a more sophisticated mixing solution with a magnetic field acting upon the paramagnetic extraction beads directly added to the RT-LAMP reaction.

Contamination from the massive amount of amplicons produced by LAMP, (10–20 μg [30]) presents an often overlooked risk of false positive results. For example, so-called “CRISPR-based”, but in actuality LAMP-based, methods utilizing lateral flow dipsticks for generalized testing make no reference in their publication [10] to this imminent threat of workspace contamination. The ultimate sealing of a finalized RT-LAMP reaction with layers of wax acts as an additional safeguard against cross-contamination of samples.

The colorimetric readout of end-point LAMP reactions dependent on pH change is a method that has gained much attention in providing easy and fast testing solutions to the COVID-19 pandemic with numerous publications replicating and optimizing protocols [31,32], including a recently-released research-use-only SARS-CoV-2 detection master mix from New England Biolabs (E2019S). The drawback of pH-based visualization is that it requires careful and standardized sampling and extraction processes upstream of RT-LAMP in order to keep the starting pH consistent with the colorimetric read-out conditions. Considering that real-world samples from humans vary in pH and that one of the principal strengths of LAMP is the robustness of Bst DNA Polymerase, Large Fragment in the face of wide pH ranges and environmental contaminants, constraining the reaction to such specific conditions is counter-productive. QUASR, DARQ [33], or other sequence-specific reporting of amplicons such as one-step strand displacement (OSD) [34] break the reliance on the delicate pH balance of the reaction, provide more accuracy with fewer false positives arising from spurious non-specific amplification products, and allow the multiplexing of reactions.

Despite using the sequence-specific detection system QUASR, a number of false positive results were encountered in the development of the ALERT platform. Testing across different SARS-CoV-2-specific primer sets, it was observed, in accordance with others [12], that while having higher sensitivity Bst 3.0 is more prone to mis-amplification and false positives in comparison to Bst 2.0. In addition to following strict guidelines of never opening a LAMP amplified tube at the same work site, choosing a sensitive primer set (such the NM set utilized in the v2 cartridges) alongside an enzyme less prone to spurious amplification (such as Bst 2.0) or having a strict cut-off of incubation times is essential to reduce false positive errors.

Some researchers have suggested the use of inexpensive, less-sensitive tests administered with high frequency as a way out of the COVID-19 pandemic [35]. This might seem like a worthy goal; however, sensitivity need not be compromised for the reduction of cost. The reagents and consumables in the ALERT system are approximately $2 per detection. Half of this price tag is composed of proprietary enzymes and several groups are actively working towards replacing them with enzymes in the public domain, such as HIV-1 reverse transcriptase [12] and non-proprietary versions of Bst LF [36]. The ALERT research group is also in the process of switching out proprietary enzymes for those in the public domain (www.reclone.org, accessed 20 October 2020). The minimal, and reusable, hardware necessary (an incubator and light source) to conduct RT-LAMP reactions with visual results, and the open-source solutions already existing such as the $2 GMO Detective fluorescence visualizer make inexpensive tests a distinct possibility without compromising on the sensitivity delivered by RT-PCR. In terms of an incubation heat source, several groups have successfully run LAMP reactions with a number of creative solutions ranging from Sous Vide [37] to thermochemical solutions such as pocket warmers [38].

## 5. Conclusions

COVID-ALERT is the first DIY, inexpensive (<$5), shipped-at-ambient temperature test for SARS-CoV-2, developed to help combat the COVID-19 pandemic. It allows a person to self-swab, isolate viral RNA, perform RT-LAMP reaction, and visualize the results all within 60 min in a simple 5-step protocol.

To enable sample lysis, both proteinase K as well as TCEP/EDTA were employed with TCEP/EDTA found to have greater sensitivity and needing only a brief 95 °C incubation. Following lysis, in order to isolate RNA, the paramagnetic bead protocol from Kellner et al. (2020) was followed. However, to remove pipetting and elution steps and to ultimately enable a DIY test, a novel magnetic wand design was introduced. This wand can simply be inserted into a tube with lysed sample and RNA-binding beads, and the paramagnetic beads are pulled out and directly added to the RT-LAMP reaction with no training.

ALERT’s central reaction is RT-LAMP with QUASR reporting for detection of viral RNA, reducing the possibility of false positives and allowing for multiplexing applications. We demonstrate, following other QUASR-LAMP publications [39], that this mode of signal generation is compatible with at least two targets in the same closed tube (BPIFA1 and SARS-CoV-2 or SARS-CoV-2 and influenza B). A simple light-emitting diode (LED) excitation source and a colored plastic gel filter allows easy discrimination between positive and negative QUASR signals.

Finally, but perhaps most crucially, the need for cold storage was removed by utilizing a variety of low melting point waxes to produce reaction-ready cartridges, stable at room temperature for at least 1 month, and potentially up to 3 months.

The COVID-19 pandemic has shaken up the field of molecular diagnostics with the drastic increase in demand for RT-PCR reactions. Although LAMP has been around for two decades, the current pandemic has introduced its value to a whole new community of researchers and technologists. This is underscored by our assay being one of multiple RT-LAMP-based assays to advance to the 20-team finals of the Rapid Covid Testing XPRIZE competition. The tools that are being developed are not only valuable for containing the current pandemic but also for future ones that are sure to emerge and other pathogens already affecting large populations yet receiving scant attention. Additionally, these tools also have applications beyond public health, including agriculture and supply chain quality assurance. Perhaps the democratization and decentralization of molecular diagnostics will be the silver lining emerging from the COVID-19 pandemic.

## Figures and Tables

**Figure 1 viruses-13-00742-f001:**
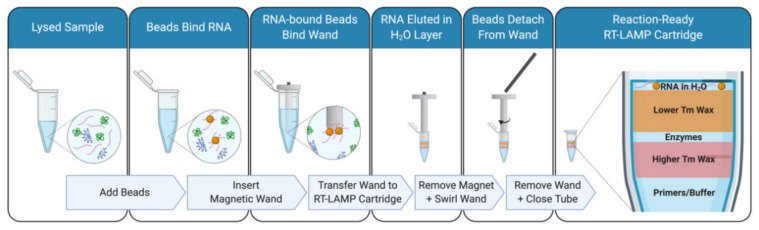
Sample concentration with magnetic wand and paramagnetic wand. The paramagnetic beads bind to viral RNA in the lysed sample. The magnetic wand is used (1) to collect these RNA-bound beads away from the rest of the sample volume and (2) to transfer them to the loop-mediated isothermal amplification (LAMP) reaction cartridge. To release the beads from the magnetic wand, the magnet is slid from the 3D-printed sheath. In the absence of a magnetic field, the paramagnetic beads are no longer magnetized and are able go into solution in water at the top of the LAMP reaction cartridge.

**Figure 2 viruses-13-00742-f002:**
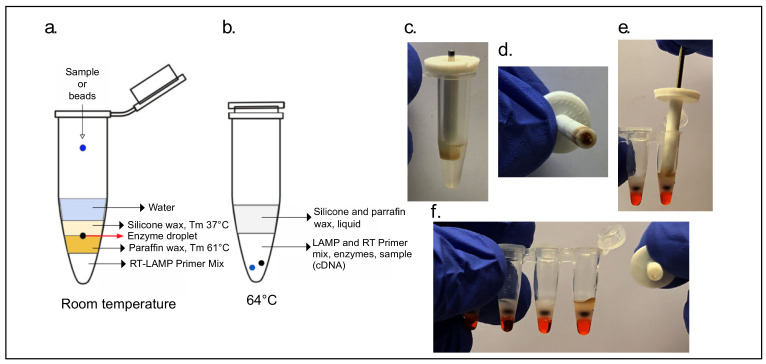
Easy to use Accessible LAMP-Enabled Rapid Test (ALERT) cartridges. One-step reaction-ready cartridge layering: cartridge layers at (**a**) room temperature and (**b**) at 64 °C. Demonstration of magnetic wand; (**c**,**d**) wand accumulates beads and (**e**) is placed into the reaction cartridge where (**f**) the magnet is removed to release beads, reagents have dyes added for demonstration purposes: primer mix is orange and the enzyme droplet is black.

**Figure 3 viruses-13-00742-f003:**
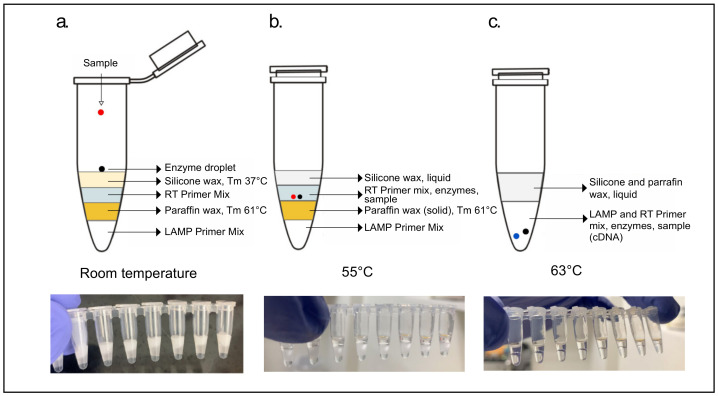
Two-step reaction-ready cartridge layering. Cartridge layers at (**a**) room temperature, (**b**) 55 °C and (**c**) at 63 °C.

**Figure 4 viruses-13-00742-f004:**
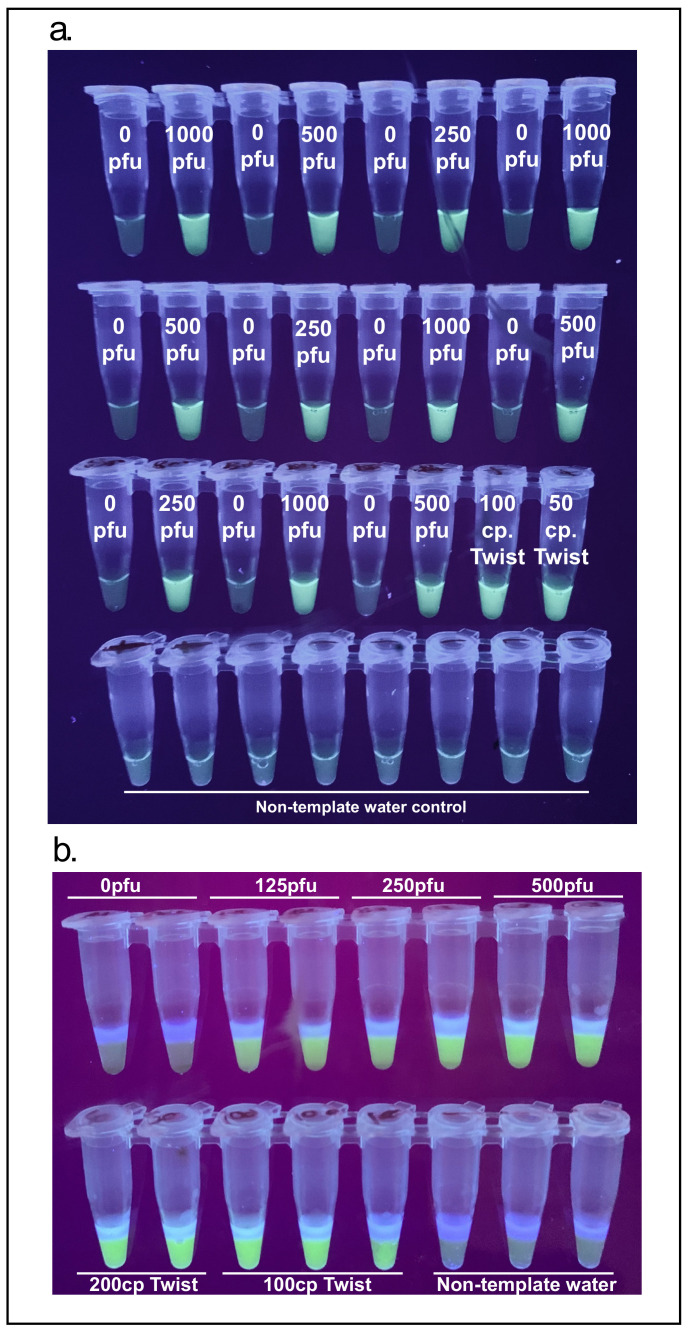
(**a**) Proteinase K lysis efficacy on nasal mid-turbinate (NMT) swabs spiked with 0, 1000, 500 and 250 pfus of severe acute respiratory syndrome coronavirus 2 (SARS-CoV-2), 100 copies of synthetic SARS-CoV-2 RNA, 50 copies of synthetic SARS-CoV-2 RNA and non-template water controls.; (**b**) TCEP/EDTA lysis efficacy on NMT swabs spiked with inactivated SARS-CoV-2. Bottom strip shows Twist Biosciences SARS-CoV-2 RNA and negative water controls.

**Figure 5 viruses-13-00742-f005:**
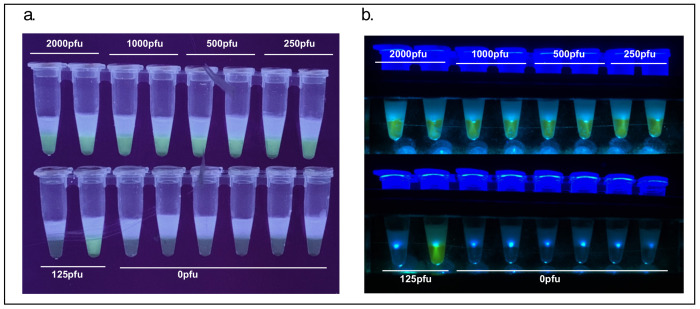
(**a**) Detection of SARS-CoV-2 using proteinase K, magnetic wand extraction and a one-step cartridge, NMT swabs spiked with inactivated SARS-CoV-2; (**b**) same tubes visualized with the GMO Detective.

**Figure 6 viruses-13-00742-f006:**
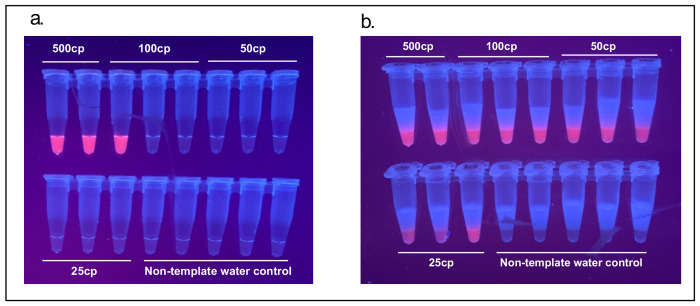
LoD improvement with a 2-step system (**a**) 1-Step RT-LAMP limit of detection (LoD) confirmed to be at ~500 copies of Twist Biosciences SARS-CoV-2 control as reported by Zhang et al. 2020; (**b**) 2-step RT/LAMP cartridge with increased sensitivity with LoD down to 25 copies of Twist Biosciences SARS-CoV-2 control.

**Figure 7 viruses-13-00742-f007:**
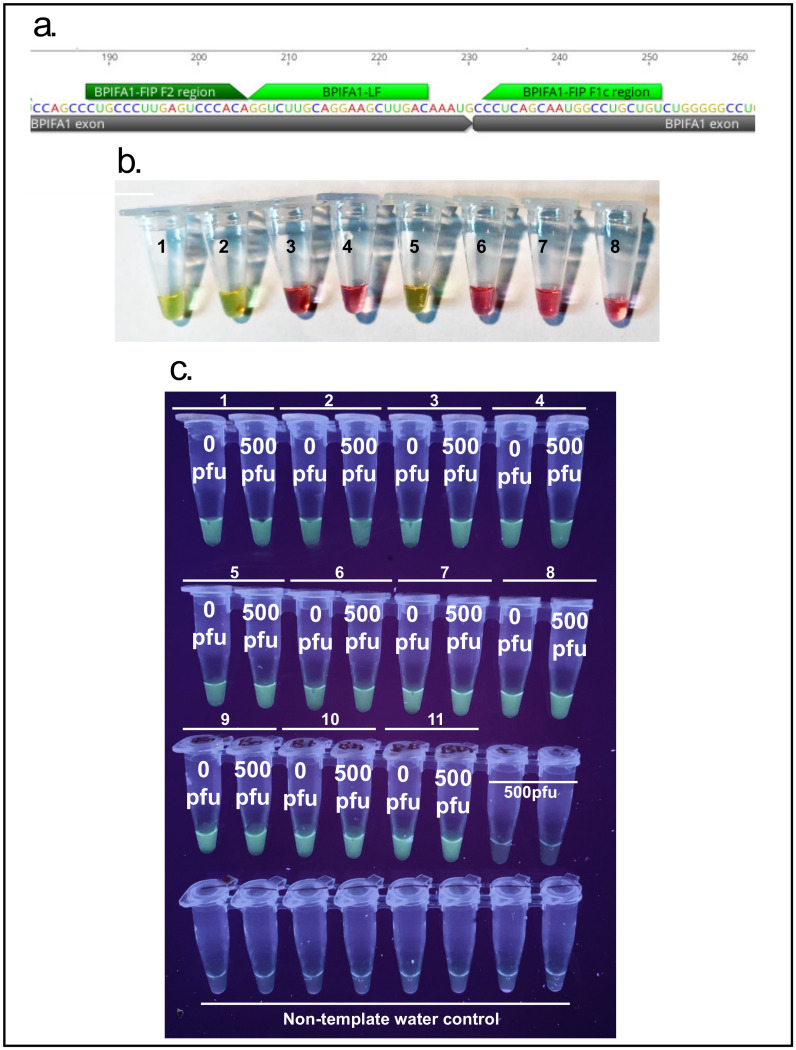
(**a**) BPIFA1 Primers were designed to traverse exon-exon junctions to be mRNA specific; (**b**) BPIFA1 RT-LAMP primer set specificity using New England Biolabs Colorimetric RT-LAMP kit: 1. Nasopharyngeal swab, TRIzol/chloroform extraction 2. Nasopharyngeal swab, Qiagen Viral RNA Mini kit extraction 3. Saliva RNA 4. Blood RNA 5. Nasopharynx flush-through (NFT), Qiagen Viral RNA Mini Kit extraction 6. Human (male) gDNA 7,8. Non-template water control, (**c**)—BPIFA1 detection across 11 individuals spiked and non-spiked with SARS-Cov-2, 500 PFU of SARS-CoV-2 and non-template water controls.

**Figure 8 viruses-13-00742-f008:**
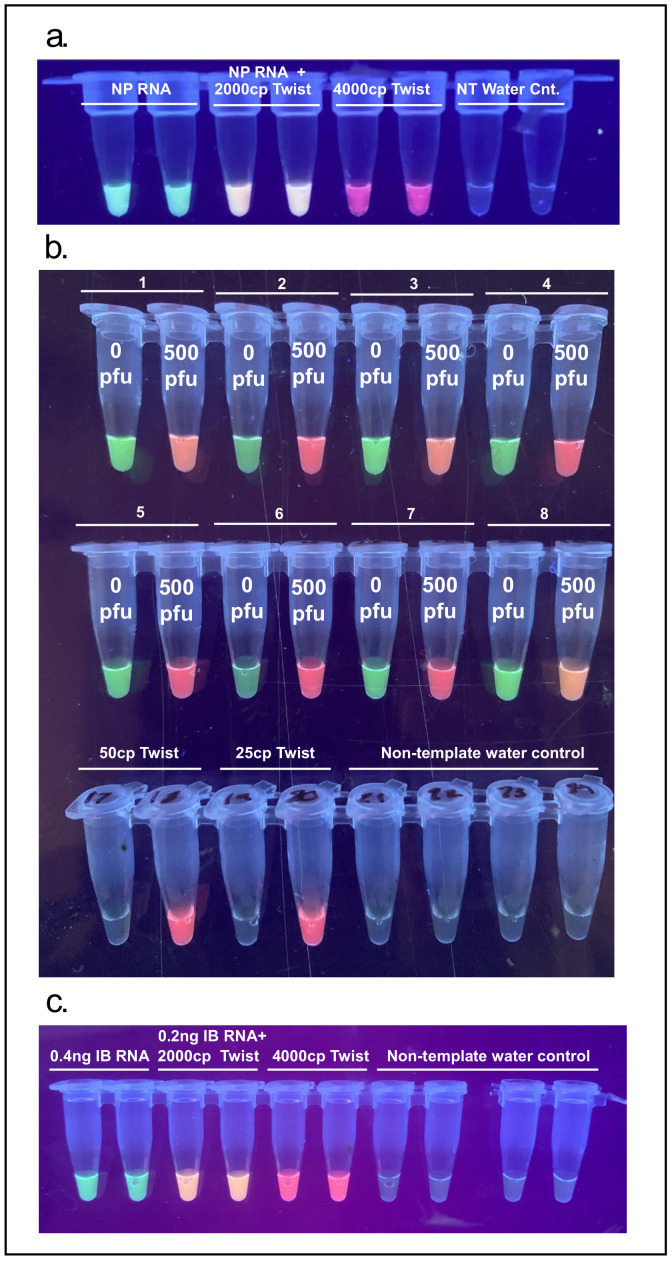
Multiplexed reactions allow for the detection of different viral targets or internal controls: (**a**) green fluorescence indicates BPIFA1 amplification, red fluorescence indicates Twist Bioscience SARS-CoV-2 control amplification, and orange fluorescence indicates both BPIFA1 and Twist Bioscience SARS-CoV-2 control amplification, (NP—nasopharynx); (**b**) multiplexed reactions for BPIFA1 and SARS-CoV-2, 11 Nasal pharynx flush thru (NFT) samples and also spiked with 500 PFU of SARS-CoV-2 and extracted with TRIzol chloroform method, 50 cp Twist synthetic controls, 25 cp Twist synthetic controls and non-template water controls; (**c**) green fluorescence indicates influenza B (IB) RNA amplification, red fluorescence indicates SARS-CoV-2 amplification, and orange fluorescence indicates both influenza B and SARS-CoV-2 amplification.

**Figure 9 viruses-13-00742-f009:**
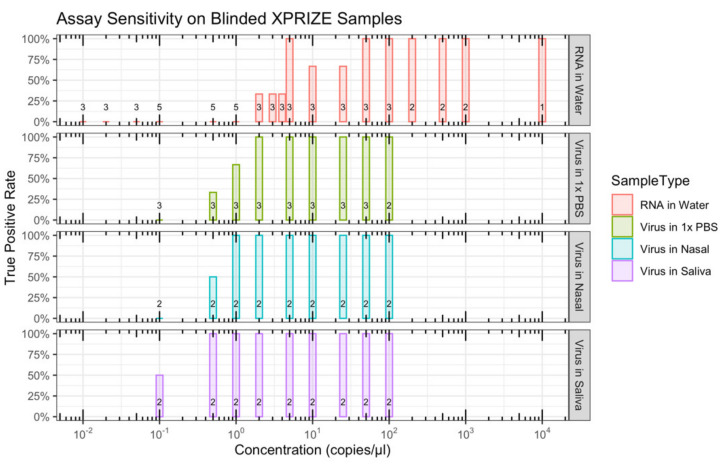
Assay sensitivity on blinded XPRIZE samples: The set of blinded XPRIZE samples comprised 123 samples with an unknown concentration SARS-CoV-2 Twist synthetic RNAs in water or SARS-CoV-2 ZeptoMetrix particles in phosphate-buffered saline (PBS), nasal sample, or saliva sample. True positive rate (*y*-axis) is shown for different concentrations (*x*-axis) across the different matrices. The number of samples tested at each concentration for a sample type is represented by the values above the *x*-axes.

**Table 1 viruses-13-00742-t001:** Limits of detection (LOD) for a 95% true positive rate based on results from testing blinded XPRIZE samples.

	Approximate LOD (Copies/uL) for 95% TPR
RNA (water)	25–50
Virus (any)	1–2
Virus (1 × PBS)	1–2
Virus (nasal)	0.5–1
Virus (saliva)	0.1–0.5

**Table 2 viruses-13-00742-t002:** True negative rates based on results from testing blinded XPRIZE samples.

	Positive Result	*n*	True Negative Rate
Mock Test Samples	0	6	100.0%
Cross-Reactivity Test Samples	1	30	96.7%
Mock + Cross-Reactivity Test Samples	1	36	97.2

## Data Availability

Not applicable.

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
