# Peer review of "Accessible LAMP-Enabled Rapid Test (ALERT) for Detecting SARS-CoV-2"

_viruses, 2021, doi:10.3390/v13050742_

Round 1
Reviewer 1 Report
Labeling of figures:
use same case for all figures throughout text. Some of the figure references are out of order. Please relabel so that the manuscript is easier to follow.
for eg: Fig 1a-b is followed by fig 2a-c &fig 3 before fig2 d-e, fig 4a,b and finally the text refers to fig 1g,h. Similarly text refers to 6c, 6a & 6b in that order
Figure 1: Legends for (d) and (e) should be corrected to (e) and (f)
Figure 2: legend for 2E should read (e)
Fig 5a must be cited in text
Supplementary Table 1:
- Specify the different fluors used here: i.e BHQ1:Black Hole Quencher 1, IAbRQSp : Iowa Black RQ-Sp, IABkFQ : Iowa Black FQ; TxRed: Texas Red
- While legend refers to "*' and "**", these are not indicated in the table itself.
Author Response
Dear Reviewer,
Thank you for these extremely helpful suggestions on how to improve our manuscript.
COMMENT: -Labeling of figures:
use same case for all figures throughout text. Some of the figure references are out of order. Please relabel so that the manuscript is easier to follow.
for eg: Fig 1a-b is followed by fig 2a-c &fig 3 before fig2 d-e, fig 4a,b and finally the text refers to fig 1g,h. Similarly text refers to 6c, 6a & 6b in that order
ANSWER: The figures have been organized to be presented in a chronological order throughout the text.
COMMENT: Figure 1: Legends for (d) and (e) should be corrected to (e) and (f)
ANSWER: Corrected
COMMENT: Figure 2: legend for 2E should read (e)
ANSWER: Corrected
COMMENT: Fig 5a must be cited in text
ANSWER: It is now cited in section 3.5
COMMENT: Supplementary Table 1:
- Specify the different fluors used here: i.e BHQ1:Black Hole Quencher 1, IAbRQSp : Iowa Black RQ-Sp, IABkFQ : Iowa Black FQ; TxRed: Texas Red
- While legend refers to "*' and "**", these are not indicated in the table itself.
ANSWER: We have included a legend for the modifications done to the oligo at the end of the table.There is an “*” by BPIFA1 and “**” by LF-TX from the SARS-CoV-2 NA/NB set.
Reviewer 2 Report
Dear Author,
This manuscript is very well written, however, in numerous sections the language used is too informal and needs to be of a more passive impersonal style. For example, the term “in our hands” used in line 404 is too informal. Also, there is far too much use of the term “we” throughout the manuscript.
The outline of the protocols used in this manuscript are very detailed, however, in the opinion of this reviewer these procedural outlines could be more concise in parts so as to improve how the article reads.
The title of section 3.6 in the results needs consideration. The term multiplexing is used here, however, in the previous related section of the methods the term duplexing is used. There should be a more direct connection between section headings in the methods and results sections.
In the abstract, the terms “increased shelf-life”, “improved specificity” and “increased sensitivity” are used, however, unless these terms are used in a direct comparative way, such as, “improved specificity compared to PCR” etc, these terms are pointless and misleading.
The figures in this manuscript need attention. The text in Figure 1A and B is difficult to read, and the quality of pictures used in Figure 1C-F could be improved. The text in Figure 2A-C is difficult to read. Figure 3 gives a good description of the technology workflow, and it is the opinion of this reviewer that this image should be introduced earlier in the manuscript, prior to the current Figure 1, so as to help the reader better understand the overall method before showing the actual wand setup. Figure 4 is very difficult to follow and needs consideration, perhaps direct labelling of the various tubes to indicate what each reaction contains would be better. Figure 5, again, the same issue as with Figure 4, hard to follow, perhaps a table could be used to help summarise the results? Figure 6, same issue as Figures 4 and 5.
The setup of this method using wax layers etc seems very laborious. How practical is this method in terms of commercialisation or rolling out on a larger scale compared to for example lyophilisation?
Author Response
Dear Reviewer,
Thank you for these very helpful comments on how to improve our manuscript. We have incorporated your suggestions throughout the text and figures.
COMMENT:-This manuscript is very well written, however, in numerous sections the language used is too informal and needs to be of a more passive impersonal style. For example, the term “in our hands” used in line 404 is too informal. Also, there is far too much use of the term “we” throughout the manuscript.
ANSWER: All but 5 of the “we”s have been removed for a more passive impersonal style. “In our hands” has also been removed. Similar edits have been made to remove the word “our”.
COMMENT: -The outline of the protocols used in this manuscript are very detailed, however, in the opinion of this reviewer these procedural outlines could be more concise in parts so as to improve how the article reads.
ANSWER: Several different iterations of how to present the protocolwas tried but considering the various components and sub-components of the reaction cartridges we were not able to make it any more concise. The figures presented in the methods section should help guide readers along on how to replicate our cartridges.
COMMENT:-The title of section 3.6 in the results needs consideration. The term multiplexing is used here, however, in the previous related section of the methods the term duplexing is used. There should be a more direct connection between section headings in the methods and results sections.
ANSWER: Multiplexing is now used throughout the manuscript to provide more consistency.
COMMENT:-In the abstract, the terms “increased shelf-life”, “improved specificity” and “increased sensitivity” are used, however, unless these terms are used in a direct comparative way, such as, “improved specificity compared to PCR” etc, these terms are pointless and misleading.
ANSWER: These claims have been qualified in our abstract.
COMMENT:-The figures in this manuscript need attention. The text in Figure 1A and B is difficult to read, and the quality of pictures used in Figure 1C-F could be improved. The text in Figure 2A-C is difficult to read. Figure 3 gives a good description of the technology workflow, and it is the opinion of this reviewer that this image should be introduced earlier in the manuscript, prior to the current Figure 1, so as to help the reader better understand the overall method before showing the actual wand setup. Figure 4 is very difficult to follow and needs consideration, perhaps direct labelling of the various tubes to indicate what each reaction contains would be better. Figure 5, again, the same issue as with Figure 4, hard to follow, perhaps a table could be used to help summarise the results? Figure 6, same issue as Figures 4 and 5.
ANSWER: Thank you for these helpful suggestions. Figure 3 has been moved to the top as Figure 1. Figure 2 and 3 have been split into two figures each in order to provide a more logical flow throughout the manuscript. The difficult to read text has been replaced to be more legible and the quality of the pictures in figure 2 (now 3) has been improved. The individual reactions in figure 4, 5 and 6 (now 5, 8 and 9) have been labelled appropriately and the captions edited to reflect this. It was chosen not to integrate all results into a single table since many different combinations of primers and extraction methodologies are presented in the results.
COMMENT:-The setup of this method using wax layers etc seems very laborious. How practical is this method in terms of commercialisation or rolling out on a larger scale compared to for example lyophilisation?
ANSWER: The sentence starting at line 749 acknowledges lyophilization methods. A further sentence was added to illustrate the feasibility of high throughput manufacturing with waxes:
“Although freeze-drying and drying methods are currently more suitable for high-throughput manufacturing, the waxes employed in the ALERT system are available in mm scale solid pellets, allowing for automatic deposition followed by a brief heating step to melt the wax, and seal reagents. “
Once again, thank you very much for the care and time to review our manuscript.